# ArfGAP with Dual Pleckstrin Homology Domains 2 Promotes Hypertrophy of Cultured Neonatal Cardiomyocytes

**DOI:** 10.3390/ijms26157588

**Published:** 2025-08-06

**Authors:** Jonathan Berthiaume, Audrey-Ann Dumont, Lauralyne Dumont, Marie-Frédérique Roy, Hugo Giguère, Mannix Auger-Messier

**Affiliations:** 1Département de Médecine—Service de Cardiologie, Centre de Recherche du CHUS, Faculté de Médecine et des Sciences de la Santé, Université de Sherbrooke, Sherbrooke, QC J1H 5N4, Canada; jonathan.berthiaume@usherbrooke.ca (J.B.); audrey-ann.dumont@usherbrooke.ca (A.-A.D.); lauralyne.dumont@usherbrooke.ca (L.D.); marie-frederique.roy@usherbrooke.ca (M.-F.R.); hugo.giguere@usherbrooke.ca (H.G.); 2Institut de Pharmacologie de Sherbrooke, Université de Sherbrooke, Sherbrooke, QC J1H 5N4, Canada

**Keywords:** Adap1, Adap2, cardiomyocyte, Centaurin-α, hypertrophy, tubulin detyrosination

## Abstract

Cardiomyocyte hypertrophy is regulated by several factors, including the ADP-ribosylation factor (Arf) family of small G proteins, among others. For instance, ArfGAP with dual pleckstrin homology domains 1 (Adap1) exerts an anti-hypertrophic effect in cultured cardiomyocytes. Its homologous protein, Adap2, is also expressed in the heart but its role remains elusive. To elucidate its function, we investigated the effects of adenoviral-mediated overexpression of Adap2 in cultured neonatal rat ventricular myocytes under both basal and pro-hypertrophic conditions, employing a range of microscopy and biochemical techniques. Despite minimal detection in neonatal rat hearts, Adap2 was found to be well expressed in adult rat hearts, being predominantly localized at the membrane fraction. In contrast to Adap1, overexpression of Adap2 provokes the robust accumulation of β1-integrin at the cellular surface of cultured cardiomyocytes. Interestingly, overexpressed Adap2 relocalizes at the sarcolemma and increases the size of cardiomyocytes upon phenylephrine stimulation, despite attenuating Erk1/2 phosphorylation and *Nppa* gene expression. Under these same conditions, cardiomyocytes overexpressing Adap2 also express higher level of detyrosinated tubulin, a marker of hypertrophic response. These findings provide new insights into the pro-hypertrophic function of Adap2 in cardiomyocytes.

## 1. Introduction

ArfGAP with dual pleckstrin homology (PH) domains 1 (Adap1, originally referred to as Centaurin-α1) was previously shown to prevent cell spreading and hypertrophic remodeling of neonatal rat ventricular myocytes (NRVMs) in culture [1]. Although Adap1 shares the same overall protein structure with its close homolog Adap2, they only exhibit a 59% protein sequence identity [2]. This likely explains their distinct binding selectivity toward different phosphoinositides, such as PI(3,4)P2, PI(4,5)P2, phosphatidylinositol (3,4,5)-trisphosphate (PIP3), and Ins(1,3,4,5)P4 through their C-terminal PH domains. These differences also suggest they could exert different roles in cell functions [2,3]. Furthermore, their expression pattern differs across various organs. While Adap1 is predominantly enriched in the brain [4,5,6], Adap2 is expressed at similar levels in several tissues of adult rats, including adipose tissue, brain, skeletal, and cardiac muscle [2]. Adap2 was also detected in the embryonic hearts of zebrafish, mouse, and human, with notable expression levels during key phases of cardiac development and throughout the later embryonic stages [7,8].

Interestingly, Adap2 was reported to regulate the actin cytoskeleton via Arf6, which is involved in cell motility and the surface expression of integral proteins [9]. Adap proteins regulate the activity of Arf6 via their GAP domain [9,10]. The Arf family comprises six proteins that are not entirely redundant in their functions, as each has specific subcellular localizations [11]. Arfs are monomeric small GTP-binding proteins ubiquitously expressed and act as molecular switches that cycle between an active GTP-bound and an inactive GDP-bound form [12]. Members of this family are critical regulators of many cellular processes, ranging from cell proliferation, polarization, adhesion, motility, to intracellular membrane trafficking and cytoskeleton protein reorganization, such as cortical actin and microtubules [9,13,14]. The latter process is involved in β1-integrins trafficking and transportation during cell spreading [15,16], which is vital for cell adhesion and cell migration [17]. Accordingly, β1-integrins impact the hypertrophic response [18] and cell spreading [19] of cultured cardiomyocytes. This aligns with our previous findings, revealing the impact of Adap1 on cultured cardiomyocytes [1]. This study found that Adap1 inhibits the cellular spreading of cardiomyocytes in culture under pro-hypertrophic stimuli. This inhibition occurs by reducing the expression of β1-integrin at the surface of cardiomyocytes, which is essential in regulating the hypertrophic and cardiac contraction/relaxation processes [20].

To gain a deeper understanding of the cardiac functions of Adap2, this study sheds light on the role of Adap2 within cultured NRVMs. Since Adap2 has been identified to act on cytoskeleton organization, and Adap1 is involved in β1-integrin modulation in cardiomyocytes, we sought to determine the impact of Adap2 on cardiomyocyte cell spreading upon stimulation with the pro-hypertrophic α-adrenergic agonist phenylephrine. Our findings show that Adap2 exerts a pro-hypertrophic effect on NRVMs.

## 2. Results

### 2.1. Adap2 Is Predominantly Localized, Unlike Adap1, at the Membranes of Adult Rat Hearts

Although initially identified two decades ago in the entire heart of adult Wistar rats [2], little is known about the role of Adap2 in cardiac cells. Hence, we assessed the mRNA and protein expression levels of Adap2 in both the entire heart and isolated cardiomyocytes from Sprague Dawley rats. Compared to neonatal rats, the relative expression level of *Adap2* mRNA is significantly increased by 3.8-fold in the hearts of adult rats (Figure 1A). In these same tissues, the relative mRNA expression levels of the Adap1 homolog remained unchanged (Figure 1A). Detection by western blotting of Adap2 protein in isolated cardiomyocytes corroborated its increased gene expression in the heart in adulthood (Figure 1B). Indeed, the Adap2 protein is barely detectable in neonatal rat ventricular myocytes (NRVMs), but its expression is clearly augmented in adult rat ventricular myocytes.

Given that Adap1 was abundantly found in the cytoplasm [4] and Adap2 was exclusively detected in the membranes [2] of different rat tissues, we sought to confirm their respective localization in the sarcoplasm and total membrane fractions from whole rat hearts. Western blot analysis of fractionated neonatal and adult rat hearts confirmed the significant enrichment of Adap2 in the membrane fraction of adult hearts, as opposed to Adap1, which is predominantly detected in the sarcoplasm of those same hearts (Figure 1C,D). These results confirm that the homologous Adap1 and Adap2 are differentially localized in cardiac cells, suggesting that they might exert divergent effects in cardiomyocytes.

### 2.2. Adap2 Increases β1-Integrin Cell Surface Localization

Overexpression of Adap1 was shown to limit cardiomyocyte hypertrophy in culture by reducing β1-integrin cell surface localization [1]. Therefore, we investigated whether adenoviral-mediated overexpression of Adap2 in NRVMs could also modify the abundance of β1-integrin present at the sarcolemma. Efficient overexpression of Adap1 or Adap2 in cultured NRVMs was confirmed by western blotting (Figure 2A). Using a cell-surface biotinylation assay coupled with western blotting, we measured the abundance of β1-integrin at the sarcolemma relative to its total expression in NRVMs overexpressing control β-Galactosidase, Adap1, or Adap2 (Figure 2B,C). As expected from previous findings [1], the amount of β1-integrin present at the cell surface of NRVMs tended to be reduced by Adap1 overexpression. Interestingly, Adap2 overexpression exerted an opposite effect and significantly increased by 2-fold, as compared to control cells, the amount of β1-integrin localized at the sarcolemma (Figure 2C).

### 2.3. Phenylephrine Stimulation Enhances Adap2 Localization at the Sarcolemma and Promotes Cell Spreading of Cultured Cardiomyocytes

Increased β1-integrin expression potentiates α1-adrenergic receptor-mediated physiological hypertrophy of cardiomyocytes [18]. Therefore, we next sought to evaluate if the overexpression of Adap2 would affect the size of NRVMs upon pro-hypertrophic stimulation with phenylephrine (PE), a specific agonist of α1-adrenergic receptor. Additionally, the cellular localization of Adap2 under basal and PE stimulation was examined. Previously, the overexpression of Adap2 was documented to localize in the cytoplasm of cells under serum starvation [9]. However, it can predominantly relocate to the plasma membrane in response to stimuli, efficiently impeding Arf6 activity and influencing actin remodeling in various cell types, notably PC12, COS-7, HeLa, and HEK293 cells [9]. Immunofluorescence performed on serum-deprived cardiomyocytes revealed the diffuse localization of overexpressed Adap2 to the sarcoplasm and the apparent accumulation at the sarcolemma of quiescent NRVMs (Figure 3A). In these same conditions, overexpressed Adap1 is predominantly localized at the sarcoplasm of NRVMs and seems to significantly colocalize with the actin filaments. Moreover, Adap1 also appears to accumulate, at least partly, and unlike Adap2, at the peri-nuclear and nuclear compartments of most NRVMs. Most strikingly, PE stimulation provokes a clear shift in the localization of Adap2 toward the sarcolemma (Figure 3A). Conversely, Adap1 stayed mainly localized at uncharacterized sub-compartments of the sarcoplasm, yet distinct from the actin filaments, under identical stimulation conditions.

Interestingly, NRVMs overexpressing Adap2 and stimulated with PE appeared to be markedly larger than control cells, suggesting an impact of Adap2 on cardiomyocyte hypertrophy and cell spreading. We quantified this observation with the Operetta high-content analysis system and by measuring the cells’ size under basal and PE-stimulated conditions (Figure 3B,C). This method systematically measures the surface of every individual α-actinin-stained NRVM detected in the view plan of each respective condition. As opposed to the significantly reduced cell surface area of Adap1 overexpressing NRVMs, consistent with previous findings [1], Adap2 overexpression promotes a modest increase in cell size of quiescent cardiomyocytes relative to control NRVMs overexpressing β-Galactosidase (Figure 3C). Furthermore, Adap2 potentiates the effect of PE pro-hypertrophic stimuli by increasing the cell size of NRVMs when compared to β-Gal condition. Upon PE stimulation, the difference in cell size between Adap1- and Adap2-overexpressing NRVMs is also amplified. Therefore, unlike Adap1, Adap2 does not prevent the cell spreading of cardiomyocytes but instead seems to act in synergy with the PE-α1-adrenergic receptor signaling to amplify the hypertrophic response of cardiomyocytes. Hence, our results support the idea of a distinct function of Adap2, as compared with Adap1, which would lead to cell spreading of cardiomyocytes during hypertrophy by increasing β1-integrin cell surface expression.

### 2.4. Adap2 Attenuates Part of the Phenylephrine-Induced Hypertrophic Signaling

Stimulation of the MAP kinase Erk1/2 pathways and reactivation of the fetal gene program are associated with PE-induced cardiomyocyte hypertrophy [21,22]. As Adap2 increases the cell size of PE-stimulated NRVMs, we assessed whether this effect can be attributed to the synergistic activation of Erk1/2 pro-hypertrophic signaling and transcriptional changes. Interestingly, Adap2 partially impedes Erk1/2 phosphorylation following PE stimulation (Figure 4A,B), suggesting its pro-hypertrophic action on NRVMs does not critically rely on this MAP kinase signaling pathway. Under these same conditions, Erk1/2 phosphorylation is unchanged by Adap1 overexpression (Figure 4A,B), as shown before [1]. As expected, PE stimulation reduces the *Myh7* mRNA expression level and increases the *Nppa* mRNA transcript abundance in NRVMs (Figure 4C). These transcriptional changes are related to cardiomyocyte hypertrophy [23]. Compared to β-Gal control, overexpression of Adap1 and Adap2 further reduces the *Myh7* mRNA expression in PE-stimulated NRVMs (Figure 4C). This is contrasting with the smaller increase in *Nppa* mRNA expression observed in these same cells, although not significantly reduced in the case of Adap2 (Figure 4C). The discrepancy between the cell-spreading effect of Adap2 and these signaling events associated with cardiomyocyte hypertrophy suggests that Adap2 may influence other dominant cellular processes, potentially explaining the increased cell size upon its expression in cardiomyocytes.

### 2.5. Adap2 Increases the Detyrosinated Tubulin Levels in Cardiomyocytes

Adap2 has been shown to interact with β-tubulin, contributing to the stabilization of microtubules through an enhancement of post-translational modifications of tubulin [24]. Detyrosination is among the several modifications impacting microtubules, and this process is heightened during PE-induced cardiomyocyte hypertrophy [25,26]. Given that Adap2 interacts with β-tubulin and influences the spreading of cardiomyocytes, we aimed to assess whether Adap2 affects the levels of detyrosinated tubulin in NRVMs. Like previous findings [25,26], we also observed by western blot an increase in the level of detyrosinated tubulin, although not significant, in PE-stimulated β-Galactosidase control NRVMs. Interestingly, Adap2 overexpression in these cells more than doubled the detyrosinated tubulin levels as compared to control conditions in both basal and PE-stimulated NRVMs (Figure 5A,B). Furthermore, Adap1 did not influence the levels of detyrosinated tubulin in either condition (Figure 5A,B).

Tubulin detyrosination generates the formation of more stable polymerized microtubules [27,28,29] and provokes an alteration of the contractile function in patients with heart failure [30]. Hence, confocal microscopy was performed to assess whether the elevation of tubulin detyrosination resulting from Adap2 overexpression correlates with an expansion of microtubule structures in NRVMs. Immunofluorescence analysis showed the prominent formation of polymerized microtubules in cells overexpressing Adap2 under basal conditions, which was further enhanced by PE stimulation (Figure 5C). Contrastingly, Adap1 did not appear to influence the organization of detyrosinated microtubules, either with or without PE stimulation. Future studies are needed to verify if these in vitro findings translate into similar observations in adult cardiomyocytes of the whole heart.

## 3. Discussion

Physiological hypertrophy of the heart is accompanied by the increased expression of many important proteins required for the normal functions and maturation of cardiomyocytes during postnatal development [31,32]. Interestingly, Adap2 is well expressed in cardiomyocytes of young adult rats. Indeed, our results show that Adap2 is not abundantly expressed in rat cardiomyocytes a few days after birth, but becomes easily detected in these contractile cells from young adult rats. This observation is also supported by another group investigating the transcript of postnatal mouse heart development [33]. They noticed that *Adap2* mRNA in the mouse whole heart progressively increased up to 1.6-fold at postnatal day 23 compared to day 1. Whether Adap2 is necessary for physiological hypertrophy during postnatal heart growth or if it contributes to pathological remodeling in cardiac diseases remains unknown but could be addressed in the future with, for example, an Adap2 conditional knockout mouse model.

By sharing a certain proportion of their sequences and PH domain, the latter being the most common domain associated with GAP proteins [34], both Adap1 and Adap2 can bind PIP3 and PIP2 [5,9]. However, their localization is highly distinct in basal conditions. We showed that it is also true in the whole heart of Sprague Dawley rats. Our results about the membrane localization of Adap2 in rat heart corroborated previous results [2], whereas Adap1 is mainly found in the cytoplasm of cardiac cells, as would be expected from its localization in other cell types [5,10]. Rapid translocation from the cytosol to the plasma membrane of cells was shown for both Adap1 and Adap2 in culture. To our knowledge, the impact of sustained stimulation on their respective localization has never been evaluated before. PE chronic stimulation over 48 h showed that Adap2, unlike Adap1, preferentially localized at the sarcolemma of NRVMs. Interestingly, this localization also corresponds to the cellular compartment where Adap2 is found in the adult heart. However, we cannot exclude that Adap2 enrichment in the total membranes from the whole heart is not due to the concomitant enrichment of microtubules in the same fraction. Precise examination of its localization by immunocytochemistry in intact adult cardiomyocytes with a refined and reproducible approach is needed to answer this unaddressed question [35]. 

Belonging to the same family group, Adap1 and Adap2 could have similar functions due to their similar primary sequences. However, their opposite effect on the cell spreading of cultured cardiomyocytes suggests that they are not redundant. Indeed, while Adap1 reduces cell size, Adap2 increases the cell surface of NRVMs. Furthermore, in the Adap2 overexpression condition with PE stimulation, several protrusions per cardiomyocyte cultured on plastic are generated at the cell membrane. These protrusions could be produced by actin cytoskeleton reorganization. Several ArfGAP domain-containing proteins, such as Asap1 [36,37], Arap1 [38], and Adap1 [39], are known to modulate effectors impacting the actin cytoskeleton organization [40]. Primarily located at the plasma membrane and endosomal compartment of many cell types, Arf6 is one of these effectors regulated by Adap2 that orchestrate the cytoskeleton dynamics during cell spreading [9,41]. More precisely, Arf6 was shown to modulate cortical actin [9,14] and microtubules [15]. Thus, the increased cell size observed with overexpression of Adap2 may be mostly due to more extensive cell spreading via protuberance of the cytoskeleton, and not to pro-hypertrophic mechanisms per se.

Guanine exchange factors (GEFs) and GAPs regulate the activity of Arf6 through their respective action on the cycling between the GTP-binding activated state and the inactivated GDP-binding state [42]. This mechanism leads to the recycling of endosomal proteins to the plasma membrane, such as β1-integrins, whichare also involved in cell spreading [43]. Interestingly, Adap1 and Adap2 again have an opposite effect on the β1-integrins recycling towards the cell surface. Since Arf6 is a known regulator of integrin cell surface recycling [43], it is plausible that Adap2 inhibits Arf6 activity at the sarcolemma of cardiomyocytes and impedes integrin recycling to increase its surface expression.

Finally, our findings show that tubulin detyrosination is facilitated through the overexpression of Adap2 and PE stimulation. Since the increase in tubulin detyrosination correlates with enhanced stiffness of cardiomyocytes [44], it would be interesting to evaluate how Adap2 alters this mechanism in pathological heart models. Indeed, cardiomyocytes from diseased hearts, such as rat myocardial infarction [45] and human heart failure [30], display elevated levels of tubulin detyrosination, and show signs of increased stiffness. In addition, cardiomyocytes treated with parthenolide to reduce tubulin detyrosination exhibit a more flexible and healthier phenotype than untreated cells [45].

Overall, our study brings to light novel information about Adap2 in cardiomyocyte spreading and hypertrophy. Novel in vivo models altering the expression or function of Adap2 are needed to expand our knowledge about its function in the whole heart, both in physiological and pathological conditions.

## 4. Materials and Methods

### 4.1. Animals and Experimental Protocols

Timed pregnant Sprague Dawley rats (strain code 001) were purchased from Charles River Laboratories (Saint-Constant, QC, Canada) and maintained in a temperature-controlled room with a 12 h light/dark cycle at the animal facility of the Université de Sherbrooke. All animal procedures performed in this study were approved by our institutional ethical committee (protocol #2017-2061), in compliance with the policies and directives of the Canadian Council on Animal Care and with the ARRIVE guidelines.

### 4.2. Rat Heart Fractionation and Subcellular Enrichment

Cellular fractionation of neonatal and adult rat hearts was performed to enrich the cytoplasm and membrane fractions, as described previously [2]. Briefly, the hearts from three neonatal (1 to 3-day-old) rats were pooled together as a single replicate, and one adult (4-month-old) rat was collected separately, and the atria were removed by excision. Hearts were then cut into small pieces, placed in 6 mL of HES buffer (20 mM HEPES, pH 7.2, 1 mM EDTA, 255 mM sucrose) supplemented with 1× Halt Protease and Phosphatase Inhibitor Cocktail (PI78447, Thermo Fisher Scientific, Mississauga, ON, Canada), and homogenized with a probe sonicator (power 5, 3 cycles of 20 s). An aliquot of 200 µL (Total fraction) was collected and frozen at −80 °C for further analysis. Homogenates were then centrifuged at 1000× *g* at 4 °C for 2 min. The supernatant containing the cytoplasm and membranes fractions was collected and placed in 5.6 mL ultracentrifugation tubes (Beckman Coulter, Mississauga, ON, Canada). Tubes were centrifuged at 55,000× *g* at 4 °C for 30 min. The supernatant (Cytoplasm fraction) was aliquoted and frozen at −80 °C for further analysis. The pellet (Membranes fraction) was resuspended in 3 mL of HES buffer, aliquoted, and stored at −80 °C for further analysis.

### 4.3. Adenovirus Generation

The adenovirus encoding β-Galactosidase (adβ-Gal) was a generous gift from Jeffery D. Molkentin (Howard Hughes Medical Institute, Cincinnati, OH, USA). The generation of 3XFLAG-tagged hADAP1 and 3XFLAG-mAdap2 was performed by subcloning their respective cDNA into the Gateway pENTR3C Dual selection vector (Invitrogen, Waltham, MA, USA) and recombining them in pAd/CMV/V5-DEST adenovirus vector (Invitrogen). The plasmids were linearized using PacI enzyme (New England Biolabs, Ipswich, MA, USA), and 4 µg of linearized plasmid was transfected into HEK293A cells (Thermo Fisher Scientific, Mississauga, ON, Canada) using Lipofectamine 2000 transfection reagent (Invitrogen). Eight to nine days post-transfection, the adenoviruses were collected, aliquoted following three freeze-thaw cycles and a centrifugation step (100× *g*, 5 min) to remove debris, and preserved at −80 °C. HEK293A cells forming a monolayer were infected with serial dilutions of each adenovirus, cultured for 48 h, methanol-fixed, and then incubated with an anti-hexon antibody (ab8249, Abcam, Waltham, MA, USA) to determine their respective titer by immunofluorescence.

### 4.4. Adult Rat Ventricular Myocytes Isolation

Adult rat ventricular myocytes (ARVMs) were isolated from 3 to 4-month-old Sprague Dawley rats. Each rat was anesthetized (1 L/min with 3% isoflurane), and the heart was collected in ice-cold Tyrode solution supplemented with 30 mM 2,3-Butanedione monoxime (BDM, Sigma-Aldrich, Oakville, ON, Canada) and 2 mM EGTA. The heart was cleaned in a sterile petri dish to remove non-cardiac tissue, and the aorta was cannulated on a Langendorff setup. The heart was then perfused with Tyrode/BDM/EGTA solution for 10 min, followed by a perfusion with Tyrode/BDM solution supplemented with 200 µM CaCl_2_ and 50,000 U of Collagenase Type II (220–280 U/mg, low trypsin, Worthington Biochemical Corporation, Lakewood, NJ, USA) for 8–12 min. Atria were removed from the digested heart, and ventricles were minced in a beaker containing 15 mL of fresh Tyrode/BDM/Collagenase/CaCl_2_ solution supplemented with 0.25% BSA and stirred for 3 min. The supernatant was collected, filtered with sterile gauze in a 15 mL conical tube, and the process was repeated with a fresh solution volume of 15 mL. Cells were washed twice with Tyrode/BDM/CaCl_2_ solution supplemented with 0.5% BSA before protein extraction in RIPA lysis buffer (50 mM Tris-HCl, pH 7.4, 150 mM NaCl, 1% Triton X-100, 1% sodium deoxycholate, 0.1% SDS, 1 mM DTT, 5 mM EDTA, and 1× Halt Protease and Phosphatase Inhibitor Cocktail) for western blot analysis.

### 4.5. Neonatal Rat Ventricular Myocytes Culture

Neonatal rat ventricular cardiomyocytes (NRVMs) were isolated and cultured using the Neonatal Cardiomyocyte Isolation System (Worthington Biochemical Corporation, Lakewood, NJ, USA). Briefly, hearts from neonatal (2 to 3-day-old) rats were collected in Hank’s Balanced Salt Solution and digested overnight at 4 °C with Trypsin (50 µg/mL, Worthington Biochemical Corporation). Following inhibition with the soybean trypsin inhibitor (200 µg/mL, Worthington Biochemical Corporation) and further digestion with collagenase (100 U/mL, Worthington Biochemical Corporation), hearts were triturated and filtered with a 70 µm cell strainer. Following centrifugation and pellet resuspension, cells were pre-plated for 30 min on 10 cm uncoated petri dishes to remove excess fibroblasts. Enriched NRVMs not yet adhered were recovered and then plated in M199 medium (supplemented with 10% fetal bovine serum (FBS), 2 mM glutamine, 100 U/mL of penicillin, and 100 μg/mL of streptomycin) at a density of 86,000 cells/cm^2^ for mRNA/protein extraction and confocal immunofluorescence experiments or in 96-well plates at 10,000 cells/well for cell surface area measurements on the Operetta high-content analysis system (Software Columbus 2.5.1, PerkinElmer, Woodbridge, ON, Canada).

### 4.6. Quantitative PCR and mRNA Expression Analysis

Expression levels of mRNA in whole heart and cultured cardiomyocytes were evaluated by quantitative reverse transcription PCR (RT-qPCR). First, the RNeasy Mini Kit (Qiagen, Toronto, ON, Canada) and the Aurum^TM^ RNA Mini Kit (Bio-Rad, Mississauga, ON, Canada) commercial kits were used, according to the manufacturer’s protocol, to extract mRNA from whole heart samples and cardiomyocytes, respectively. The iScript^TM^ Reverse Transcription Supermix for RT-qPCR (Bio-Rad) was then used to synthesize corresponding cDNA. Diluted cDNA products were then used for qPCR reaction using validated primers (listed in Table 1), the SsoAdvanced^TM^ Universal SYBR Green Supermix (Bio-Rad), and a Mastercycler RealPlex (Eppendorf, Mississauga, ON, Canada). Gene expression analysis was then performed using the 2^(−ΔΔCt)^ method [46], relative to the Rpl30 housekeeping gene.

### 4.7. Protein Expression Analysis

Proteins were extracted from NRVMs 72 h post-infection or ARVMs freshly isolated with RIPA lysis buffer. The DC Protein assay kit (Bio-Rad) was used to quantify protein concentration from each lysate. An equal amount of diluted proteins was migrated on SDS-PAGE gels and transferred to nitrocellulose membranes (GE Healthcare, Mississauga, ON, Canada). Western blots were performed using the following antibodies: mouse anti-Adap2 (H00055803-B01, Abnova, Taipei City, Taiwan); rabbit anti-Adap1 (ABS179) and rabbit anti-Detyrosinated-tubulin (AB3201) were purchased from EMD Millipore (Burlington, MA, USA); rabbit anti-Gapdh (14C10), rabbit anti-α-tubulin (2144S), anti-rabbit HRP-linked (7074S) and anti-mouse HRP-linked (7076S) were purchased from Cell Signaling (Danvers, MA, USA); rabbit anti-β1-integrin (sc-8978) and mouse anti-Cadherin (sc-1499) were purchased from Santa-Cruz Biotechnology (Dallas, TX, USA); mouse anti-α-actinin (A7811) and anti-goat HRP-linked (A5420) were purchased from Sigma. Chemiluminescent signal was detected using the Luminata Western HRP Crescendo ECL Substrate (EMD Millipore). Acquisition and data analysis were performed using the ChemiDoc MP system and Image Lab software version 5.2 from Bio-Rad.

### 4.8. Confocal Microscopy and Immunofluorescence Analysis

For confocal microscopy imaging using the Olympus Fluo View FV1000 system (40× magnification), NRVMs were fixed with 4% paraformaldehyde, permeabilized with ice-cold 20% [*v*/*v*] methanol for 10 min, and stained with 4′,6-diamidino-2-phenylindole (DAPI; D9542, Sigma-Aldrich, Oakville, ON, Canada), phalloidin-iFluor 647 (ab176759) and some of the antibodies mentioned above. Image analysis was performed using the Olympus FV10 ASW 4.0 Viewer analysis software. Cell surface area measurements and immunofluorescence analysis using the Operetta high-content analysis system (PerkinElmer) were performed as previously described [1]. NRVMs were infected with adenoviruses (multiplicity of infection = 50) and cultured for 8 h in M199 medium supplemented with 10% FBS before being rinsed and cultured for another 48 h with serum-free M199 medium supplemented or not with phenylephrine (PE, 50 µM).

### 4.9. Surface β1-Integrin Pull-Down Assay

Proteins expressed at the surface of NRVMs were labeled with biotin for pull-down assay as previously described [1]. Briefly, cells were rinsed three times with phosphate buffered saline (PBS)-Ca^2+^/Mg^2+^ (100 mg/mL CaCl_2_, 100 mg/mL MgCl_2_ 6H_2_O, pH 8.0) and incubated for 1 h at 4 °C with Sulfo-NHS-SS-Biotin (A8005, APExBIO) diluted in PBS-Ca^2+/^Mg^2+^ Afterwards, cells were rinsed once with PBS-Ca^2+/^Mg^2+^ containing 100 mM glycine and twice with PBS-Ca^2+/^Mg^2+^. Proteins were extracted in RIPA lysis buffer, and their concentration was determined with the DC Protein assay kit (Bio-Rad). An aliquot of 100 µg of proteins (Total fraction) was collected for further analysis. Another aliquot of proteins (600 µg) was incubated with MagReSyn Streptavidin microsphere beads (ReSyn Biosciences, Halifax, NS, Canada) for 2 h at 4 °C. Biotinylated proteins were then eluted in boiling Laemmli loading buffer for 10 min and migrated on SDS-PAGE gel for cell-surface β1-integrin expression analysis by western blot.

### 4.10. Statistical Analysis

Statistical analysis was performed on all experiments from at least three independent replicates, using GraphPad Prism 8 software. All experiment units were included, and none were excluded from the experimental group (no inclusion/exclusion criteria were established a priori). One-way and two-way ANOVA analyses were performed using multi-parametric testing when appropriate, as indicated in figure legends. Data are expressed as mean ± standard deviation, and *p*-value < 0.05 was considered significantly different.

## Figures and Tables

**Figure 1 ijms-26-07588-f001:**
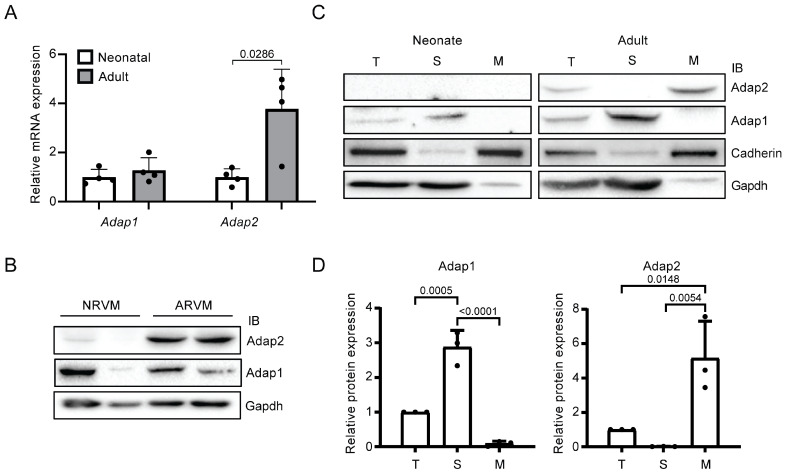
Cardiac expression of homologous Adap proteins in rats. (**A**), Relative *Adap1* and *Adap2* mRNA expression levels in neonatal (2- to 3-day-old) and young adult (3- to 4-month-old) Sprague Dawley rat hearts measured by RT-qPCR and normalized to the *Rpl30* reporter gene (*n* = 4 independent isolations). (**B**), Representative western blots showing Adap1, Adap2, and Gapdh (loading control) protein expression in neonatal rat ventricular myocytes (NRVM) compared to adult rat ventricular myocytes (ARVM) (*n* = 2 independent isolations). (**C**), Representative western blots showing the subcellular localization of Adap1, Adap2, Cadherin (membrane marker), and Gapdh (sarcoplasm marker) proteins from fractionated neonatal and young adult rat cardiac ventricles (T: Total, S: Sarcoplasm, M: Membranes). (**D**), Densitometric analysis of Adap1 and Adap2 protein expression in the corresponding subcellular fractions from young adult rat cardiac ventricles, as shown in (**C**), and relative to the total extract (*n* = 3 independent fractionations).

**Figure 2 ijms-26-07588-f002:**
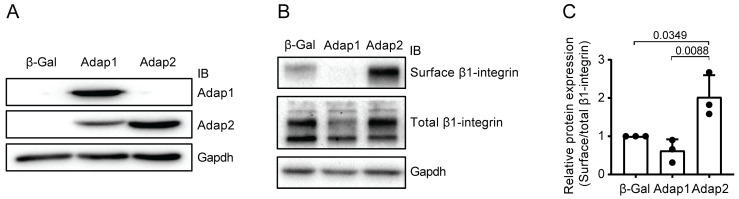
Impact of overexpressing Adap homologous proteins on cell surface expression of β1-integrin in cultured cardiomyocytes. (**A**), Representative western blots of Adap1 and Adap2 overexpression 72 h after neonatal rat ventricular myocytes (NRVMs) infection with control β-Galactosidase (β-Gal), Adap1, and Adap2 adenoviruses (multiplicity of infection = 50), respectively. (**B**), Representative western blots showing the cell surface expression of β1-integrin in NRVMs overexpressing either β-Gal, Adap1, or Adap2. (**C**), Densitometric analysis of β1-integrin cell surface expression relative to total β1-integrin expression in the corresponding conditions, as shown in (**B**) (*n* = 3 independent experiments).

**Figure 3 ijms-26-07588-f003:**
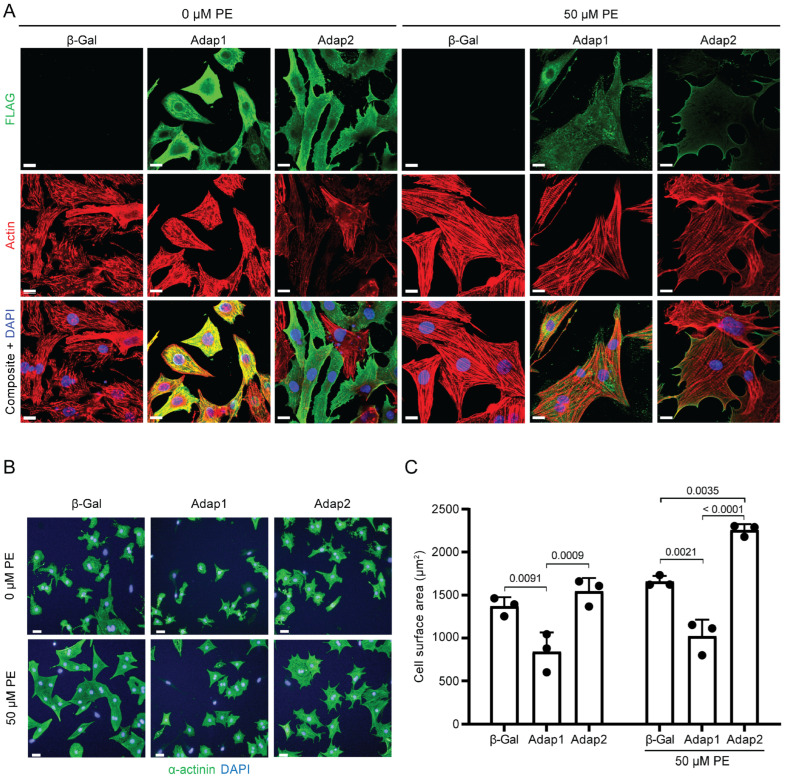
Adap2 potentiates phenylephrine-induced cardiomyocyte remodeling. (**A**), Representative confocal immunofluorescence images showing the cellular localization, 72 h after infection with adenoviruses, of endogenous actin and overexpressed 3xFLAG-ADAP1 and 3xFLAG-Adap2 in neonatal rat ventricular myocytes (NRVMs) incubated with or without phenylephrine (PE, 50 µM) for 48 h (Scale bar = 10 µm). Nuclei and actin filaments were stained with DAPI (blue) and phalloidin-iFluor 647 (red), respectively. (**B**), Representative immunofluorescence images of α-actinin in NRVMs overexpressing Adap1 or Adap2, as compared to control cells overexpressing β-Galactosidase (β-Gal), and incubated with or without phenylephrine (PE, 50 µM) for 48 h, and using the Operetta high-content analysis system (Perkin Elmer) (Scale bars = 50 µm). Nuclei were stained with DAPI. (**C**), Cell size measurements of segmented NRVMs from the corresponding conditions are shown in (**B**) (*n* = 3 independent experiments).

**Figure 4 ijms-26-07588-f004:**
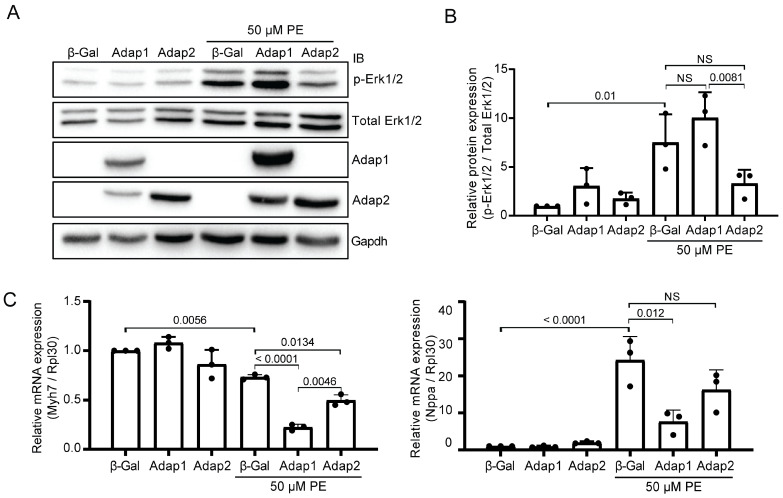
Adap2 does not amplify the phenylephrine-induced hypertrophic signaling in cardiomyocytes. (**A**), Representative western blots showing the Erk1/2 pathway activation in neonatal rat ventricular myocytes (NRVMs) overexpressing control β-Galactosidase (β-Gal), Adap1, or Adap2 and incubated with or without phenylephrine (PE, 50 µM) for 48 h. (**B**), Densitometric analysis of Erk1/2 activation in the corresponding conditions, as shown in (**A**) (*n* = 3 independent experiments). (**C**), Gene expression analysis by RT-qPCR of specific cardiac hypertrophy markers (i.e., *Myh7* and *Nppa* normalized to the *Rpl30* reporter gene) in NRVMs overexpressing control β-Gal, Adap1, or Adap2 and incubated with or without PE (50 µM) for 48 h (*n* = 3 independent experiments).

**Figure 5 ijms-26-07588-f005:**
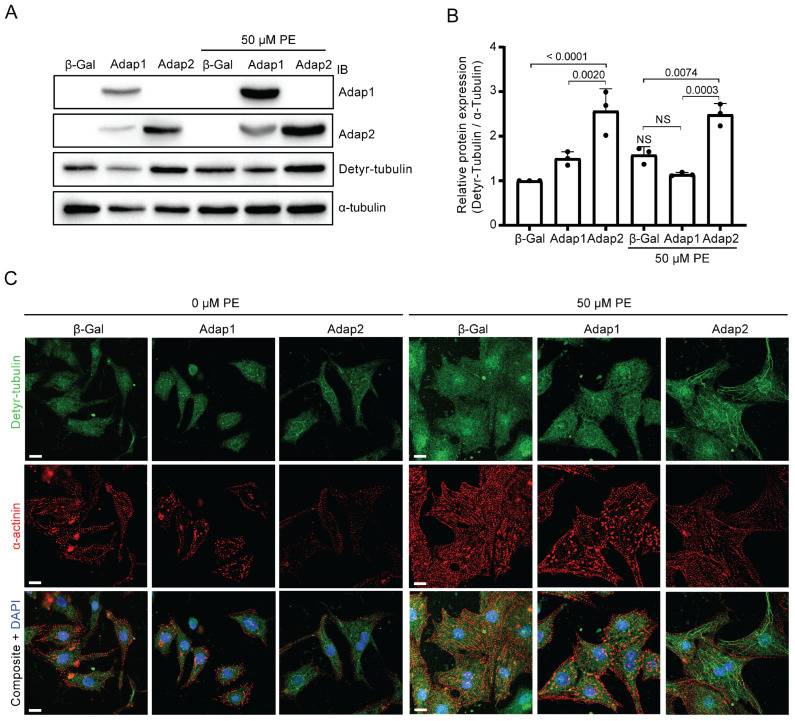
Adap2 increases microtubule detyrosination in cardiomyocytes. (**A**), Representative western blots showing the expression of detyrosinated α-tubulin (Detyr-tubulin) in neonatal rat ventricular myocytes (NRVMs) overexpressing control β-Gal, Adap1, or Adap2 and incubated with or without phenylephrine (PE, 50 µM) for 48 h. (**B**), Densitometric analysis of detyrosinated α-tubulin relative to total α-tubulin in the corresponding conditions, as shown in (**A**) (*n* = 3 independent experiments). (**C**), Representative confocal immunofluorescence images showing the detyrosinated α-tubulin from microtubule cytoskeleton and α-actinin organization in NRVMs overexpressing control β-Gal, Adap1, or Adap2 and incubated with or without PE (50 µM) for 48 h (Scale bar = 10 µm). Nuclei were stained with DAPI (blue).

**Table 1 ijms-26-07588-t001:** List of primers. Sequences of validated primers (forward: Fwd; Reverse: Rev) used for the RT-qPCR of indicated target genes.

Gene Name	Gene Description	Sequences (5′ → 3′)
*Adap1*	ArfGAP with dual PH domains 1	Fwd: CAAAGACCCTCTGGATGCCTT
		Rev: GGTGACTCTGGGTTGACGG
*Adap2*	ArfGAP with dual PH domains 2	Fwd: CATCACTCCGGAGCGGAAAT
		Rev: CAGCCAGATCCTCCGAGATG
*Myh7*	Myosin heavy chain 7	Fwd: CAACCTGTCCAAGTTCCGCA
		Rev: GGCATCCTTAGGGTTGGGTAG
*Nppa*	Natriuretic peptide A	Fwd: CGGCACTTAGCTCCCTCTCT
		Rev: GTTGCAGCCTAGTCCGCTCT
*Rpl30*	Ribosomal protein L30	Fwd: TCTTGGCGTCTGATCTTGGT
		Rev: AAGTTGGAGCCGAGAGTTGA

## Data Availability

The raw data supporting the conclusions of this article will be made available by the authors on request.

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
