# Peer review of "ArfGAP with Dual Pleckstrin Homology Domains 2 Promotes Hypertrophy of Cultured Neonatal Cardiomyocytes"

_ijms, 2025, doi:10.3390/ijms26157588_

Round 1

Reviewer 1 Report

Comments and Suggestions for Authors

In this manuscript, Berthiaume and colleagues investigated a potential role for ArfGAP With Dual Pleckstrin Homology Domains 2 (Adap2) in cardiomyocyte hypertrophy. Using gain-of-function approaches, they observe an increased hypertrophic response in NRCM with Adap2 overexpression after PE stimulation, which is opposite to what is seen with Adap1. This correlates with an increase in β1-integrin expression in the membrane fraction of these cells. Surprisingly, Adap2 overexpression attenuates the pro-hypertrophic ERK1/2 signalling pathway, despite the increase in hypertrophic response. The experiments in this manuscript are performed well, and the manuscript is well-written. I only have some minor comments to improve the manuscipt.

  1. Please show bar graphs with data points so the reader can appreciate data variability.
  2. What is the relative expression of Adap1 and 2 in these cells/the mammalian heart? Can the authors make use of (published) RNAseq data to show the relative expression of these two proteins? Also, in what cell types of the heart are these proteins expressed?
  3. The authors focus on gain-of-function of Adap2; what about loss-of-function? Is Adap2 necessary for the hypertrophic response in NRCM?
  4. Are Adap1 and Adap2 regulated in (models of) heart disease?
  5. Are the authors able to stain endogenous Adap1 and Adap2 in NRCM and ARCM?

Author Response

Reviewer 1

In this manuscript, Berthiaume and colleagues investigated a potential role for ArfGAP With Dual Pleckstrin Homology Domains 2 (Adap2) in cardiomyocyte hypertrophy. Using gain-of-function approaches, they observe an increased hypertrophic response in NRCM with Adap2 overexpression after PE stimulation, which is opposite to what is seen with Adap1. This correlates with an increase in β1-integrin expression in the membrane fraction of these cells. Surprisingly, Adap2 overexpression attenuates the pro-hypertrophic ERK1/2 signalling pathway, despite the increase in hypertrophic response. The experiments in this manuscript are performed well, and the manuscript is well-written. I only have some minor comments to improve the manuscipt.

  1. Please show bar graphs with data points so the reader can appreciate data variability.

Response: Thank you for raising this point. We now include the data points on every bar graph presented in the manuscript (i.e., Figures 1A, 1D, 2C, 3C, 4B-C, and 5B).

  1. What is the relative expression of Adap1 and 2 in these cells/the mammalian heart? Can the authors make use of (published) RNAseq data to show the relative expression of these two proteins? Also, in what cell types of the heart are these proteins expressed?

Response: According to Human Protein Atlas (PMID: 25613900), Adap1 protein is expressed at low-medium levels in the cardiomyocytes (detected by immunohistochemistry), with higher mRNA expression in the left ventricle (average nTPM: 7.9) compared to the atrium (average nTPM: 4.6). Although Adap2 protein expression in heart muscle is still pending analysis, similar expression levels of its mRNA are detected in the left ventricle and atrium (average nTPM of 3.7 and 2.8, respectively). We agree that these questions raised by the reviewer are interesting. However, they extend beyond the scope of this study, which focuses on the roles of Adap1 and Adap2 in cultured rat neonatal ventricular myocytes, rather than in other cardiac cell types. Undoubtedly, determining the relative expression of Adap1 and Adap2 in cardiac cells, as could be performed with a single-cell or single-nucleus RNAseq dataset (e.g., GSE183852 from PMID: 35959412), will lay the groundwork for further lines of investigation.

  1. The authors focus on gain-of-function of Adap2; what about loss-of-function? Is Adap2 necessary for the hypertrophic response in NRCM

Response: The expression of Adap2 is barely detectable in cultured NRVM (Fig.1), and a loss-of-function approach was not attempted in these cells. Interestingly, knockdown of ADAP2 ortholog in zebrafish with morpholino oligos revealed its importance for cardiogenesis (PMID: 24711647). Thus, future development of Adap2 conditional knock-out, as previously discussed in the manuscript (lines 309-311), will be relevant and ideal for studying its impact on cardiac physiological and pathological responses in a cell-specific manner during development and in adulthood.

  1. Are Adap1 and Adap2 regulated in (models of) heart disease?

Response: To the best of our knowledge, there is only a few evidence available regarding the regulation of ADAP1 and ADAP2 expression in models of cardiac diseases. For example, according to previously reported RNAseq data set (PMID: 34774109), there is an increase of ADAP1 and ADAP2 expression in the peripheral blood cells of cardiovascular disease patients compared to healthy individuals (ADAP1: 3.62 vs 2.72, p=0.0003; ADAP2: 1.66 vs 1.20, p=0.014; see the data in the uploaded pdf). Interestingly, ADAP2 was among three candidate genes associated with a higher incidence of congenital heart defects in individuals with neurofibromatosis type I (NF1) microdeletion syndrome (PMID: 24711647 and 16138909). While these data appear interesting, their implications for understanding the role of Adap1 and Adap2 in the context of broader heart diseases remain to be studied.

  1. Are the authors able to stain endogenous Adap1 and Adap2 in NRCM and ARCM?

Response: Upon pro-hypertrophic phenylephrine stimulation, we observed cellular localization changes of 3xFLAG-Adap1 and 3xFLAG-Adap2 within NRVMs overexpressing these proteins (Fig. 3A). However, we haven’t been successful in detecting the endogenous signals from either of these two proteins by immunofluorescence microscopy with confidence, as we lack the proper knockdown or knock-out control conditions. Staining the endogenous Adap1 and Adap2 in cardiomyocytes will also require the identification of sensitive and specific antibodies suitable for immunocytochemistry.

Reviewer 2 Report

Comments and Suggestions for Authors

Reviewer report 

ArfGAP With Dual Pleckstrin Homology Domains  Promotes Hypertrophy of Cultured Neonatal Cardiomyocytes

This is a well-designed study that illuminates novel information about Adap2 in cardiomyocyte spreading and hypertrophy. The article shows how altering the expression or function of  Adap2 is related to its function in the whole heart, both in physiological and pathological conditions. The article has scope in research design, experimental results, and editing sections.

  1. Please explain the entire review in the form of a graphical abstract with a summary and conclusion.
  2. The similarity index looks beyond 25 %, which is more than the limit. Please modify it to below 15 to 18%.
  3. Authors should add the figure that explains the pathway targeting with the pathophysiological condition mentioned in the study.
  4. Please explain the clinical co-relation of the present study.
  5. What will the effect of Pressure Overload-Induced Pathological Cardiac Hypertrophy be in the current model? Please explain.
  6. Authors should have performed an Echocardiographic assessment of the live rats to justify the hypothesis
  7. The study was performed in neonatal heart cardiomyocytes. What will be the effect on parameters if the study is performed in male rats? Please explain.

Author Response

Reviewer 2

Reviewer report

ArfGAP With Dual Pleckstrin Homology Domains Promotes Hypertrophy of Cultured Neonatal Cardiomyocytes

This is a well-designed study that illuminates novel information about Adap2 in cardiomyocyte spreading and hypertrophy. The article shows how altering the expression or function of Adap2 is related to its function in the whole heart, both in physiological and pathological conditions. The article has scope in research design, experimental results, and editing sections.

  1. Please explain the entire review in the form of a graphical abstract with a summary and conclusion.

Response: Please find in the uploaded pdf a graphical abstract generated with BioRender, summarizing the key findings of our research article. If the editor determines that it should be included in the article, we will be happy to add it.

  1. The similarity index looks beyond 25 %, which is more than the limit. Please modify it to below 15 to 18%.

Response: This similarity may have originated from Section 4.10. Statistical analysis, which was indeed like the corresponding section from our previous article about Adap1’s role in neonatal cardiomyocytes (PMID: 30206251). We modified section 4.10 (lines 435-440) in the reviewed version of the article. If other sections or sentences throughout the manuscript should be modified to reduce the similarity index, we would appreciate knowing more precisely which part they correspond to.

  1. Authors should add the figure that explains the pathway targeting with the pathophysiological condition mentioned in the study.

Response: Considering the scope of this study, which focuses on the in vitro hypertrophic effect of overexpressed Adap2 in NRVMs, it would be speculative and an overinterpretation of our results to claim that endogenously expressed Adap2 plays the same role in adult cardiomyocytes and cardiac pathophysiological conditions in vivo. This topic warrants further investigation using suitable animal models in the future. However, we generated a graphical abstract in response to comment #1, which may be included with the article and summarizes the key findings of this study.

  1. Please explain the clinical co-relation of the present study.

Response: This study investigated the in vitro effect of overexpressed Adap2 on cultured neonatal ventricular cardiomyocytes. As answered in the previous question #3, any extrapolation of our findings to in vivo clinical conditions would be an overinterpretation and potentially misleading.

  1. What will the effect of Pressure Overload-Induced Pathological Cardiac Hypertrophy be in the current model? Please explain.

Response: The rats used in this study were not exposed to any pathological stress but only served for 1) cardiomyocytes isolation and culture, or 2) whole heart fractionation for subcellular enrichment of proteins. While transverse aortic coarctation would be an interesting model to study the cardiac effects of Adap2 in pressure-overload pathological conditions, this question is beyond the scope of this study and would require novel animal models that are currently unavailable.

  1. Authors should have performed an Echocardiographic assessment of the live rats to justify the hypothesis

Response: As answered in question #5, this study only used rats to isolate cardiomyocytes for in vitro experiments. Performing in vivo echocardiographic measurements on rats would not support the tested hypothesis. While we agree that in vivo experiments with, for example, conditional knock-out mice would be interesting, it is outside the scope of the current study and should be investigated in future studies.

  1. The study was performed in neonatal heart cardiomyocytes. What will be the effect on parameters if the study is performed in male rats? Please explain.

Response: This last question raised by the reviewer, as for its previous questions #3-6, also seems to relate to the in vivo effect of Adap2. However, this is beyond the scope of our study, which identifies for the first time a role for Adap2 in cultured cardiomyocytes pooled from both male and female newborn rats without distinction. As we indicated in the discussion, we agree nonetheless that further studies and novel animal models are required to address these interesting questions. No doubt that the use of transgenic Adap2 overexpressing or conditional knock-out mice, for example, would be helpful to identify the role of Adap2 in cardiac responses to pathophysiological conditions. This will also allow studying both male and female mice to reveal if Adap2 roles in cardiomyocytes are related to any cardiovascular sex differences.
